# Effects of the Interface between Inorganic and Organic Components in a Bi_2_Te_3_–Polypyrrole Bulk Composite on Its Thermoelectric Performance

**DOI:** 10.3390/ma14113080

**Published:** 2021-06-04

**Authors:** Cham Kim, David Humberto Lopez

**Affiliations:** 1Division of Nanotechnology, Daegu Gyeongbuk Institute of Science and Technology (DGIST), 333 Techno Jungang-daero, Daegu 42988, Korea; 2Department of Chemical and Environmental Engineering, University of Arizona, 1133 E. James. E. Rogers Way, Tucson, AZ 85721, USA; davidlopez3@email.arizona.edu

**Keywords:** thermoelectric material, Bi_2_Te_3_, polypyrrole, composite, interface

## Abstract

We provided a method to hybridize Bi_2_Te_3_ with polypyrrole, thus forming an inorganic/organic bulk composite (Bi_2_Te_3_–polypyrrole), in which the effects of energy band junction and phonon scattering were expected to occur at the interface of the two components. Bi_2_Te_3_–polypyrrole exhibited a considerably high Seebeck coefficient compared to pristine Bi_2_Te_3_, and thus it recorded a somewhat increased power factor despite the loss in electrical conductivity caused by the organic component, polypyrrole. Bi_2_Te_3_–polypyrrole also exhibited much lower thermal conductivity than pristine Bi_2_Te_3_ because of the phonon scattering effect at the interface. We successfully brought about the decoupling phenomenon of electrical and thermal properties by devising an inorganic/organic composite and adjusting its fabrication condition, thereby optimizing its thermoelectric performance, which is considered the predominant property for n-type binary Bi_2_Te_3_ reported so far.

## 1. Introduction

The overall performance of thermoelectric materials is generally estimated via the dimensionless figure of merit, ZT (=*α*^2^*σT*/*κ*, *α*: Seebeck coefficient, *σ*: electrical conductivity, *κ*: the thermal conductivity, *T*: absolute temperature) [1,2,3]. A high ZT value can be achieved by increasing the electrical property—that is, the power factor (*α*^2^*σ*)—and by decreasing the thermal conductivity [4,5,6]. Many researchers have endeavored to decouple the electrical and thermal properties of diverse materials by devising nanostructures [7,8,9], by introducing nanoinclusions [10,11,12], and by injecting metallic or nonmetallic atoms [13,14,15], which were shown to increase the power factor and/or decrease the thermal conductivity. Conducting polymers are recognized as promising candidates for thermoelectric applications because of their low thermal conductivity and moderate electrical conductivity as thermoelectric materials [16,17,18]. These polymers are often offered as flexible thin films, which are easily applicable to micro energy-harvesting devices for low-temperature thermoelectric operations. Many researchers have attempted to hybridize conducting polymers with inorganic Bi_2_Te_3_ for low-temperature operations [19,20,21], as they expected to achieve low thermal conductivity due to conducting polymers and supplement electrical conductivity via inorganic Bi_2_Te_3_. However, these hybrid materials exhibited poorer electrical properties than inorganic Bi_2_Te_3_ alone, possibly due to the low carrier transport properties of conducting polymers or severe carrier scattering at the component interfaces. In the present work, we tried to decrease the possible loss in electrical properties by hybridizing inorganic Bi_2_Te_3_ with a conducting polymer (polypyrrole) and regulating the fabrication process to obtain an optimized bulk structure. We developed a brief fabrication process, in which a small amount of polypyrrole was dispersed in n-type Bi_2_Te_3_, and the resulting composite was packed via spark plasma sintering under different conditions; thus, we gained an optimized composite material in the bulk phase. The composite inevitably showed inferior electrical conductivity than pristine Bi_2_Te_3_ because of carrier scattering and carrier offsetting at the interfaces of the components, resulting in reductions in carrier mobility and carrier concentration, respectively. However, the composite recorded an improved Seebeck coefficient compared to pristine Bi_2_Te_3_ due to the decrease in the carrier concentration, thus providing a somewhat increased power factor. Due to possible phonon scattering at the interface, in addition to the reduced carrier mobility and concentration, the composite exhibited lower thermal conductivity than pristine Bi_2_Te_3_. The electrical and thermal properties were successfully decoupled and optimized, and thus, the composite material provided significantly enhanced thermoelectric performance, which is considered a predominant property for n-type binary Bi_2_Te_3_ reported so far.

## 2. Materials and Methods

### 2.1. Chemicals

Bismuth (99.99%, Vital Materials, Guangzhou, China) and tellurium granules (99.9%, Vital Materials, Guangzhou, China) with pyrrole (98%, Sigma-Aldrich, St. Louis, MO, USA) were used as raw materials. Chemical oxidation of pyrrole was conducted by using Iron(III) sulfate (97%, Sigma-Aldrich, St. Louis, MO, USA). Each chemical was used with no further purification.

### 2.2. Sample Preparation

Sample preparations were performed through our procedures described elsewhere [22]. Bi_2_Te_3_ powder was prepared by a conventional pulverization method. Proper stoichiometric amounts of the raw metal elements (Bi and Te granules) for the Bi_2_Te_3_ composite were weighed out and placed in a quartz tube, which was then evacuated and sealed. The tube was installed in a rocking furnace and heated to 900 °C (9 °C min^−1^), held at the same temperature for 24 h, and cooled to room temperature naturally. The resultant ingot was pulverized to obtain a powder. Iron(III) sulfate (50 mmol) was dissolved in distilled water (200 mL), and the Bi_2_Te_3_ powder (12.7 g) was added. To this solution, a dispersion of pyrrole (0.495 g, 7.38 mmol) in distilled water (100 mL) was added with vigorous stirring. The resulting solution was stirred for 6 h to allow the chemical oxidation of the pyrrole by iron(III) sulfate, which was expected to result in the distribution of polypyrrole throughout the Bi_2_Te_3_ powder. The solution was filtered and washed with deionized water and absolute ethanol. The resulting solid product was dried at 40 °C in a vacuum oven for 12 h, thus providing a mixed powder of Bi_2_Te_3_ with polypyrrole. The mixed powder and the pristine Bi_2_Te_3_ powder were packed using a spark plasma sintering instrument (Dr. Sinter, SPS3.20MK-IV). Approximately 12 g of each powder was separately placed into a cylindrical graphite mold (55 mm × 60 mm (*d* × *h*)) with an inner hole of a 12 mm diameter for compaction at 350 and 400 °C for 2–3 min under a pressure of 100 MPa in an Ar atmosphere, providing cylindrical bulk specimens.

### 2.3. Characterizations

#### 2.3.1. Material Characterizations

X-ray diffraction (XRD) analysis was carried out using a D/MAX-2500 diffractometer (Rigaku, Tokyo, Japan) equipped with a scintillation counter detector. The XRD patterns were obtained in a 2*θ* range of 10–80°. Fourier-transform infrared (FTIR) spectra were obtained using a continuum spectrometer (iS50, Thermo Scientific, Waltham, MA, USA, spectral range: ≤2000 cm^−1^). Raman spectra were recorded on a Nicolet spectrometer (Almeca XR, Thermo Scientific, Waltham, MA, US, excitation photon energy: 532 nm), which can cover the spectral range of 100–2000 cm^−1^. Transmission electron microscopy (TEM) with selected-area electron diffraction (SAED) was conducted using a Titan G2 instrument (Titan G2 60-300, FEI Company, Hillsboro, OR, USA).

#### 2.3.2. Thermoelectric Characterizations

The transport properties were measured for the faces of the sintered specimen that were vertical to the direction of the applied pressure. The carrier concentration (*n*) was obtained using an HMS-3000 Hall effect measurement system (Ecopia). The electrical resistivity (*ρ*) and Seebeck coefficient (*α*) were recorded in the temperature range of 25–200 °C using a ZEM-3 Seebeck/resistance measuring system (Ulvac-Rico). The carrier mobility (*μ*) was derived via the equation *μ* = 1/(*ρn*e), where *ρ* is the electrical resistivity, *n* is the carrier concentration, and e is the elementary charge. The thermal conductivity (*κ*) was calculated using the equation *κ* = *λC_p_d*, where *λ* is the thermal diffusivity, *C_p_* is the specific heat, and *d* is the physical density. The thermal diffusivity (*λ*) was recorded in the temperature range of 25–200 °C using an laser flash tool (LFA447, Netzsch, Selb, Germany). The specific heat was recorded using a differential scanning calorimeter (DSC200, Netzsch, Selb, Germany, Appendix A). The physical density (*d*) was obtained via the Archimedes method (Appendix A). The figure of merit was derived using the equation *ZT* = *α*^2^*σT*/*κ*. The thermal conductivity is roughly regarded as the sum of carrier (*κ_c_*) and lattice (*κ_l_*) thermal conductivities (i.e., *κ ≈ κ_c_* + *κ_l_*). The carrier contribution was derived via the Wiedemann–Franz law *κ_c_* = *L*_0_*σT*, where *L*_0_ is the Lorenz number (2.0 × 10^−8^ WΩK^−2^ for a degenerate semiconductor) [23], and *σ* is the electrical conductivity. The lattice contribution was obtained by subtracting the carrier contribution from the overall thermal conductivity.

## 3. Results

In situ data of the spark plasma sintering (SPS) process for the powder samples were obtained to determine appropriate sintering conditions. We monitored a profile showing the variation in the *Z*-axis displacement of a cylindrical graphite mold filled with the powder samples during the SPS process (Figure 1). The sintering temperature was elevated by adjusting the electric current, resulting in a decrease in the *Z*-axis displacement, indicative of sample compaction.

X-ray diffraction (XRD) patterns were obtained for the spark-plasma-sintered samples (Figure 2). Any noticeable change such as peak shift or new peak appearance was not detected in the diffraction patterns, although the sample was mixed with polypyrrole and/or was sintered at different temperatures.

Both FTIR and Raman spectroscopies were conducted to identify the inorganic and organic components in the samples, thus verifying whether the components were hybridized. These spectroscopies were also used to observe how the components’ molecules were changed with the sintering temperature (Figure 3).

TEM analyses were performed for the sintered samples to obtain information on morphology, crystallinity, and atomic arrangement of the inorganic and organic components. The information was used to confirm the unique interfaces between the components (Figure 4).

## 4. Discussion

We preferentially monitored a profile showing the variation in the *Z*-axis displacement of a cylindrical graphite mold filled with pristine Bi_2_Te_3_ during the SPS process (Figure 1a). As the sintering temperature was elevated by adjusting the electric current, the displacement increased because of sample compaction. When the sintering temperature reached around 350 °C, the displacement started to saturate, indicative of termination of the compaction. The sintering temperature kept increasing and then maintained at 400 °C for a few minutes to assure completion of the sintering. The resultant specimen of pristine Bi_2_Te_3_ exhibited a relative density of over 98% compared to the theoretical density (7.86 g cm^−3^) of Bi_2_Te_3_ (Appendix A, Appendix A) [24]. A similar sintering strategy was applied to the mixed power of Bi_2_Te_3_ and polypyrrole (Bi_2_Te_3_–polypyrrole). In Bi_2_Te_3_–polypyrrole, the *Z*-axis displacement began to saturate at a lower temperature by ca. 100 °C than in pristine Bi_2_T_3_ (Figure 1b) possibly due to the portion of polypyrrole having relatively poor temperature tolerance. To compare the pristine Bi_2_Te_3_ specimen, Bi_2_Te_3_–polypyrrole was preferentially sintered at the same temperature (400 °C, Appendix A, Appendix A). Considering that pristine Bi_2_Te_3_ exhibited a saturated displacement from ca. 350 °C, we also attempted to sinter Bi_2_Te_3_–polypyrrole at 350 °C (Figure 1b). Both the Bi_2_Te_3_–polypyrrole specimens sintered at the two different temperatures recorded relatively low densities compared to the pristine Bi_2_Te_3_ specimens (Appendix A, Appendix A) because of the portion of polypyrrole having much lower physical density (ca. 1.4–1.5 g cm^−3^) [25,26] than Bi_2_Te_3_.

The pristine Bi_2_Te_3_ specimens exhibited an X-ray diffraction pattern of rhombohedral Bi_2_Te_3_ (Figure 2a), identified as JCPDS Card No. 82-0358 (space group: $R\bar{3}m\;\left(D_{3d}^{5}\right)$). The Bi_2_Te_3_–polypyrrole specimens displayed identical diffraction patterns to the pristine Bi_2_Te_3_ specimens. Any diffraction pattern of polypyrrole was not observed in the Bi_2_Te_3_–polypyrrole specimens because polypyrrole exhibits only a broad diffraction peak, indicative of its typical amorphous nature (Appendix A, Appendix A). According to the enlarged diffraction plot over a 2*θ* range of 28°, no considerable peak shifts were found in the strongest diffraction peaks for the (015) plane (Figure 2b). In addition, the specimens exhibited no significant differences in lattice parameters (Appendix A, Appendix A). These results possibly indicate that the addition of polypyrrole does not cause any lattice distortion of Bi_2_Te_3_. The FTIR spectra of pristine Bi_2_Te_3_ and Bi_2_Te_3_–polypyrrole are presented in Figure 3a. The spectra of pristine Bi_2_Te_3_ showed nothing but the signal at 1384 cm^−1^, consistent with the characteristic signal of Bi_2_Te_3_ [27]. This signal seemed to overlap with one of the polypyrrole peaks in the spectra of Bi_2_Te_3_–polypyrrole. The spectra of Bi_2_Te_3_–polypyrrole exhibited various characteristic signals for polypyrrole, such as C–C stretching vibration (775, 1088, and 1165 cm^−1^), polypyrrole doping state (853, 965, and 1384 cm^−1^), C–H out-of-plane bending vibration (1039 cm^−1^), N–H deforming vibration (1243 cm^−1^), C–N stretching vibration (1288 cm^−1^), C–H in-plane stretching vibration (1321 cm^−1^), C=C stretching vibration (1448, 1543, and 1593 cm^−1^), and N–H in-plane bending vibration (1630 cm^−1^) (Figure 3b) [28,29]. Bi_2_Te_3_–polypyrrole displayed most of the IR characteristic peaks of polypyrrole. This indicates that polypyrrole was not degraded despite the high sintering temperature of Bi_2_Te_3_ (350 or 400 °C), but it retained the molecular structure possibly due to the short sintering time, thus being hybridized with Bi_2_Te_3_. Comparing the Raman spectra of the pristine Bi_2_Te_3_ and Bi_2_Te_3_–polypyrrole specimens to that of polypyrrole, it was verified that the characteristic signals of Bi_2_Te_3_ occurred below 1000 cm^−1^, while those for polypyrrole appeared above 1000 cm^−1^ (Figure 3c). The spectrum of polypyrrole clearly displayed signals corresponding to C–C or C–N ring stretching mode and C=C stretching mode at 1359 and 1578 cm^−1^, respectively (Figure 3d) [30,31]. The spectrum of polypyrrole also showed peaks at 950 and 1071 cm^−1^, which are associated with bipolaron and polaron, respectively [30]. The presence of these peaks indicates that polypyrrole is highly oxidized. These peaks noticeably disappeared in Bi_2_Te_3_–polypyrrole sintered at 350 °C (Figure 3d). In addition, Bi_2_Te_3_–polypyrrole showed a somewhat decreased peak intensity for the stretching modes in the backbone chain (i.e., C–C or C–N and C=C stretching) compared to polypyrrole. In Bi_2_Te_3_–polypyrrole sintered at 400 °C, it was confirmed that the two satellite peaks were hardly detected, and the characteristic peaks for the stretching modes in the backbone chain were reduced even more. These results indicate that the original oxidation states in polypyrrole (i.e., bipolaron and polaron) were deformed, and even the polypyrrole molecule gradually degraded as the sintering temperature increased. It is expected that the Bi_2_Te_3_–polypyrrole specimens included fewer oxidized polypyrrole molecules having relatively short backbone chains compared to pristine polypyrrole. It inevitably occurred that the polypyrrole molecules were deformed or even degraded during the sintering process, but this molecular change was considered to possibly adjust by controlling the sintering temperature. The XRD, FTIR, and Raman results consistently indicate that Bi_2_Te_3_–polypyrrole exhibits both the characteristic signals of Bi_2_Te_3_ and polypyrrole. It is expected that Bi_2_Te_3_ is not chemically bonded to polypyrrole, but the former physically generates unique interfaces with the latter. To support this conclusion, we carried out TEM analyses for the Bi_2_Te_3_–polypyrrole specimens sintered at different temperatures (Figure 4). Both the specimens were mainly composed of irregularly shaped grains with a few micrometers in size (dark contrast). The specimens also contained bright-contrast grains, which were smaller than the dark-contrast grains and were not found in the pristine Bi_2_Te_3_ specimens (Appendix A, Appendix A). The dark grains in the Bi_2_Te_3_–polypyrrole specimens were found to have distinct crystalline directions, whereas the bright grains were observed to show an amorphous-looking phase (Figure 4). Periodic atomic arrangements were clearly observed in the dark grains, where the crystal directions of Bi_2_Te_3_ were identified with a relevant SAED pattern (Figure 5a and inlet). According to the SAED pattern with the zone axis along the $\left[5\bar{5}1\right]\;$direction, the dark grains were confirmed to have lattice fringes indexed as (015) and $\left(10\bar{5}\right)$ planes with interfaces of ca. 3.2 Å. No specific atomic arrangements were found for the bright grains (Figure 4 and Figure 5), indicative of an amorphous phase of polypyrrole, and thus polypyrrole might have a diffraction peak that is too broad to appear in the Bi_2_Te_3_–polypyrrole specimens (Appendix A, Appendix A). We confirmed that the Bi_2_Te_3_ grains grew along different crystalline directions bordered by the grain boundaries with the amorphous phase of polypyrrole, resulting in the unique interfaces in the Bi_2_Te_3_–polypyrrole specimen (Figure 5b).

Typical electrical conductivity of Bi_2_Te_3_ (in the order of 10^3^ S cm^−1^) [32,33] is much higher than that of polypyrrole (in the order of 10^−2^–10^2^ S cm^−1^) [34,35] due to the predominant crystallinity of the former. Considering this great difference, it is easily comprehensible that electrical conductivity may decrease due to the polypyrrole addition to Bi_2_Te_3_. Bi_2_Te_3_–polypyrrole recorded higher electrical resistivity than pristine Bi_2_Te_3_ up to 60% (Figure 6a). For this comparison, pristine Bi_2_Te_3_ sintered at 350 °C was used because it recorded better thermoelectric properties than that sintered at 400 °C. The different thermoelectric properties of pristine Bi_2_Te_3_ are elaborated in the Appendix A (Appendix A). Bi_2_Te_3_–polypyrrole exhibited basically lower carrier concentration and mobility than pristine Bi_2_Te_3_ (Appendix A), and thus the former displayed higher electrical resistivity than the latter. In our previous work, we suggested the equilibrium band alignment with band-bending potential wells in the interface of Bi_2_Te_3_ and polypyrrole by using their energy band structures [22]. Because of the band bending, the accumulation of electrons and holes possibly occurs at the interfaces, thereby causing the carrier offsetting, which explains the decrease in the carrier concentration in Bi_2_Te_3_–polypyrrole. Carrier transport may be interrupted by the interfaces between the Bi_2_Te_3_ and polypyrrole components (Figure 5b) and by the reduced physical densities (Appendix A), resulting in the decrease in carrier mobility of Bi_2_Te_3_–polypyrrole. As the sintering temperature increased, Bi_2_Te_3_–polypyrrole exhibited increases both in the carrier concentration and mobility (Appendix A). Because of both increases, Bi_2_Te_3_–polypyrrole sintered at 400 °C recorded lower electrical resistivity than that sintered at 350 °C (Figure 6a). As verified in the Raman spectroscopy results (Figure 3d), the original oxidation states in polypyrrole (i.e., bipolaron and polaron) were deformed, and even polypyrrole degradation was accelerated as the sintering temperature increased; thus, Bi_2_Te_3_–polypyrrole sintered at 400 °C might have exhibited a higher carrier concentration and mobility than that sintered at 350 °C. As explained above (Appendix A), the carrier concentration decreased due to the addition of polypyrrole, and thus Bi_2_Te_3_–polypyrrole exhibited a higher Seebeck coefficient than pristine Bi_2_Te_3_ up to 33% (Figure 6b). When the sintering temperature increased, Bi_2_Te_3_–polypyrrole showed an increase in the carrier concentration as confirmed above, thus providing decreases in both electrical resistivity and the Seebeck coefficient (Figure 6a,b). Consequently, Bi_2_Te_3_–polypyrrole recorded a higher power factor than pristine Bi_2_Te_3_ up to 14%, whereas Bi_2_Te_3_–polypyrrole displayed no significant change in the power factor depending on the sintering temperature (Figure 6c).

Bi_2_Te_3_–polypyrrole exhibited lower thermal conductivity than pristine Bi_2_Te_3_ up to 50% because the former had lower carrier and lattice thermal conductivities than the latter (Figure 6d,e). Both the carrier concentration and mobility decreased due to the addition of polypyrrole, and thus Bi_2_Te_3_–polypyrrole showed decreased carrier thermal conductivity. In addition, Bi_2_Te_3_–polypyrrole may possess vigorous phonon scattering at the interfaces of Bi_2_Te_3_ and polypyrrole components (Figure 5b), thus providing decreased lattice thermal conductivity. However, the thermal conductivity of Bi_2_Te_3_–polypyrrole increased as the sintering temperature elevated because of the increased carrier and lattice thermal contributions (Figure 6e). The degradation of polypyrrole was accelerated due to the increased sintering temperature, and thus Bi_2_Te_3_–polypyrrole exhibited an increased carrier concentration and mobility and a decreased amount of interface for phonon scattering, resulting in the increased carrier and lattice thermal contributions, respectively.

By introducing polypyrrole to Bi_2_Te_3_ and adjusting the compaction conditions of the resulting materials, the inorganic/organic composite (Bi_2_Te_3_–polypyrrole) in the bulk phase was successfully developed. By adequately adding the organic component, the electrical property was somewhat increased, while the thermal conductivity was significantly reduced; thus, remarkable ZT enhancement was accomplished (Figure 6f). Bi_2_Te_3_–polypyrrole recorded ZT maximum (ZT_max_) and average (ZT_ave_) values of 1.18 at 100 °C and 1.12 at 50–150 °C, which were approximately twice as what pristine Bi_2_Te_3_ exhibited. These ZT values are not only superior to those previously reported as the highest performance for n-type binary Bi_2_Te_3_, but they are also as competent as the excellent performance of n-type ternary Bi_2_(Te,Se)_3_ recently reported (Appendix A, Appendix A) [36,37,38].

## 5. Conclusions

Bi_2_Te_3_ was hybridized with polypyrrole to produce a bulk composite (Bi_2_Te_3_–polypyrrole) including the interface, where the effects of energy band junction and phonon scattering effects possibly occur. The effects may cause noticeable changes in thermoelectric transport properties, as presented below.
The composite exhibited a decrease in the carrier concentration, possibly due to carrier offsetting at the interface consisting of the energy band junction. The composite showed a slight increase in the power factor compared to pristine Bi_2_Te_3_ because of the decrease in the carrier concentration.The composite recorded lower thermal conductivity than pristine Bi_2_Te_3_ because the former exhibited lower carrier and lattice thermal conductivities, which resulted from the decrease in the carrier concentration and the phonon scattering effect at the interface, respectively.The decoupling effect of the electrical and thermal properties resulted in remarkably enhanced ZT values, which may be evaluated as the highest performance among n-type binary Bi_2_Te_3_. Because of the superior ZT values at low temperatures, the composite may be highly applicable to thermoelectric operations at low temperatures, such as energy-harvesting devices and systems.

We are currently conducting further studies on the interaction mechanism and theory and various compositions between the components.

## Figures and Tables

**Figure 1 materials-14-03080-f001:**
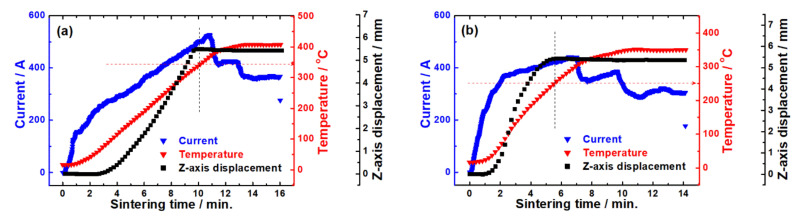
Variation in the *Z*-axis displacement of a graphite mold filled with (**a**) pristine Bi_2_Te_3_ and (**b**) Bi_2_Te_3_–polypyrrole during the spark plasma sintering process at 400 and 350 °C, respectively, accompanied by the control of the electric current.

**Figure 2 materials-14-03080-f002:**
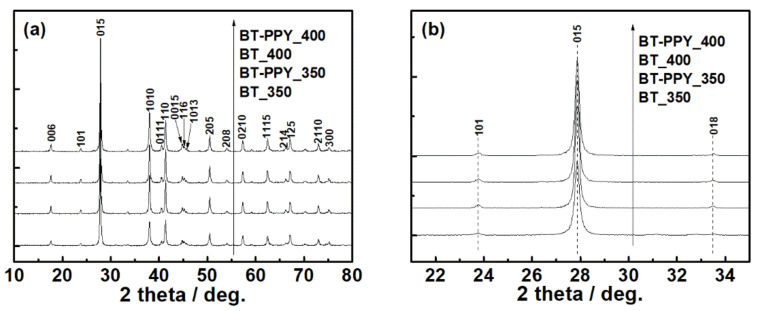
(**a**) XRD patterns of pristine Bi_2_Te_3_ (BT) and Bi_2_Te_3_–polypyrrole (BT-PPY), which were individually sintered at 350 and 400 °C; (**b**) enlarged XRD plots of the BT and BT-PPY over a 2*θ* range from 21° to 35°.

**Figure 3 materials-14-03080-f003:**
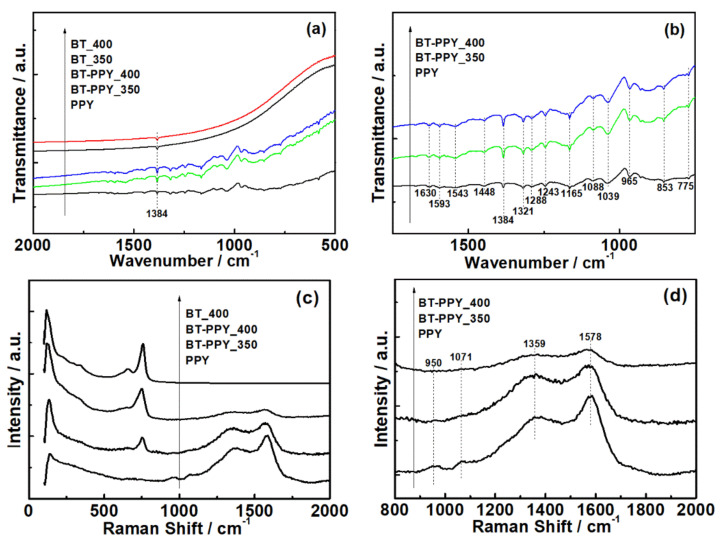
(**a**) FTIR spectra of pristine Bi_2_Te_3_ (BT) and Bi_2_Te_3_–polypyrrole (BT-PPY), which were individually sintered at 350 and 400 °C, and polypyrrole (PPY); (**b**) enlarged FTIR spectra of BT-PPY and PPY in the wavenumber range from 750 to 1750 cm^−1^; (**c**) Raman spectra of BT, BT-PPY, and PPY; (**d**) enlarged Raman spectra of BT-PPY and PPY in the range of Raman shift from 800 to 2000 cm^−1^.

**Figure 4 materials-14-03080-f004:**
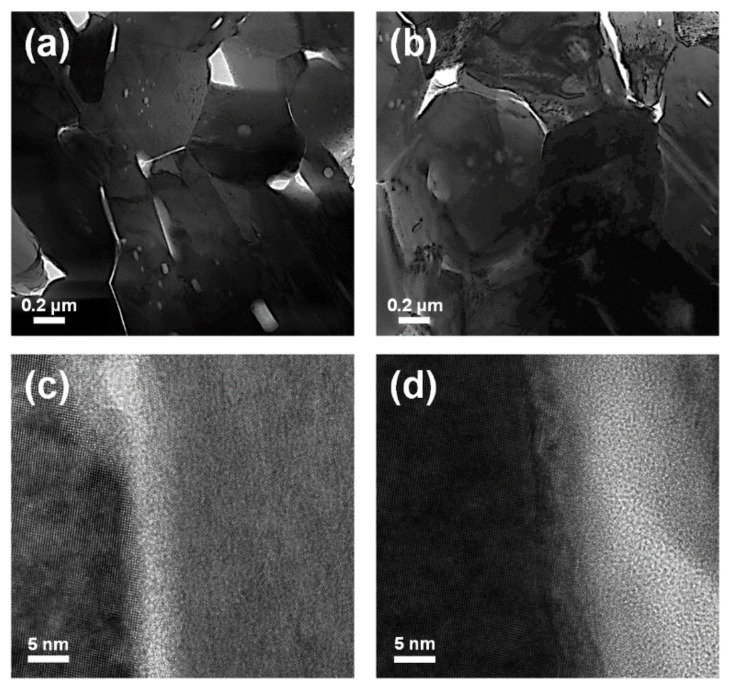
TEM images of the Bi_2_Te_3_–polypyrrole specimens sintered at 350 (**a**,**c**) and 400 °C (**b**,**d**) at different resolutions.

**Figure 5 materials-14-03080-f005:**
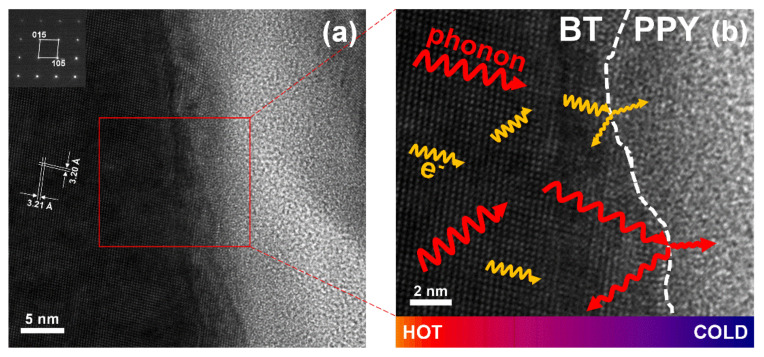
(**a**) TEM image of Bi_2_Te_3_–polypyrrole and the SAED pattern with the zone axis along the $\left[5\bar{5}1\right]$ direction (inlet) to identify the lattice fringes of Bi_2_Te_3_, (**b**) HR-TEM image of Bi_2_Te_3_–polypyrrole displaying the interface between Bi_2_Te_3_ and polypyrrole components where electron and phonon scattering may occur.

**Figure 6 materials-14-03080-f006:**
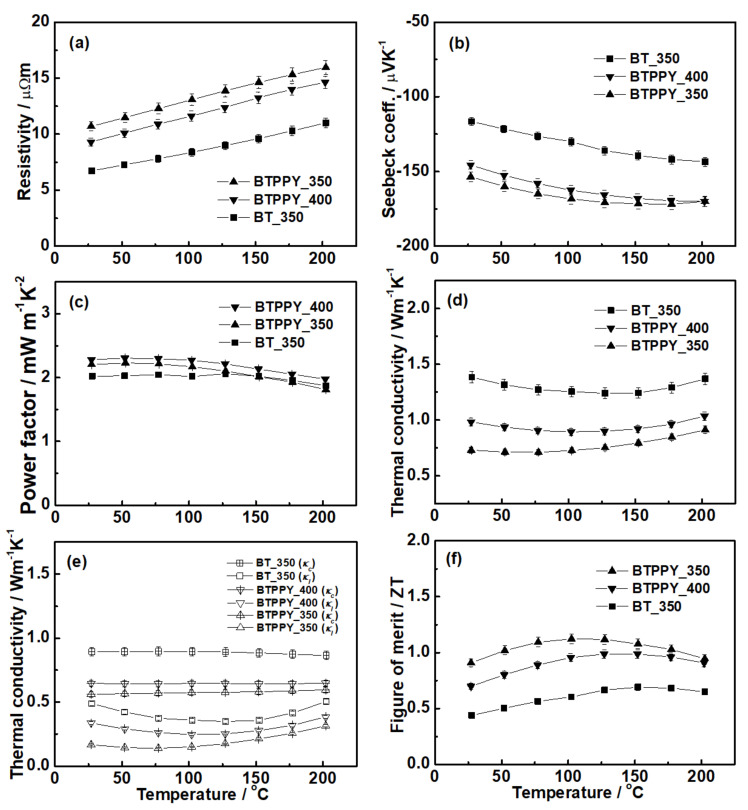
Thermoelectric properties of the pristine Bi_2_Te_3_ and Bi_2_Te_3_–polypyrrole specimens sintered at 350 and 400 °C: (**a**) electrical resistivity, (**b**) Seebeck coefficient, (**c**) power factor, (**d**) thermal conductivity divided into (**e**) carrier (*κ_c_*) and lattice (*κ_l_*) contributions, and (**f**) ZT. Error bars are shown when they exceed the symbol size.

## Data Availability

The data presented in this study are available on request from the corresponding author.

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
