# Peer review of "Effects of the Interface between Inorganic and Organic Components in a Bi2Te3–Polypyrrole Bulk Composite on Its Thermoelectric Performance"

_materials, 2021, doi:10.3390/ma14113080_

Round 1
Reviewer 1 Report
The manuscript “Effects of the interface between inorganic and organic components in a Bi2Te3-polypyrrole bulk composite on its thermoelectric performance” by C. Kim and D. Humberto Lopez reports the preparation of Bi2Te3-polypyrrole composites for thermoelectric applications This manuscript needs to be improved before publication in Materials. Major corrections are here indicated:
- A significant part of this research is included in the Supplementary Material file. However, the file has not been provided, so this work cannot be evaluated properly.
- Details about the heating and cooling rate is not included in Sample preparation section.
- Specific heat measurements are not shown in the manuscript. However, the values were used for the calculation of thermal conductivity. How is the dependence of Cp with the increment of temperature?
- Density was also needed for the calculation of thermal conductivity. A value of 98% is included in the text. However, it is not clear what is happening with the density in the composites. If there is a reduction in the density of the pellets, the decrease of thermal conductivity in the composites cannot be considered, as the samples are not comparable.
- It is weird that Section 3. Results and discussion starts with a figure.
- X ray diffraction cell parameters and space group need to be included in the text.
- Authors indicate “The Bi2Te3-polypyrrole 156 specimens displayed almost similar diffraction patterns to the pristine Bi2Te3 specimens”… almost similar? So are there differences?
- SAED patterns indexing should be checked and zone axis needs to be incorporated.
- Error bars need to be included in figure 6.
- An important number of Bi2Te3 and Bi2Te3-composites publications can be found in the literature. However, in this work there is not information about previous works and an effort of comparison between their results and literature results must be done.
Reviewer 2 Report
Title: "Effects of the interface between inorganic and organic components in a Bi2Te3-polypyrrole bulk composite on its thermoelectric performance "
In this work the authors provided the way of hybridizing Bi2Te3 with polypyrrole, thus affording the inorganic/organic bulk composite (Bi2Te3-polypyrrole), in which the effects of energy band junction and phonon scattering were expected to occur at the interface of the two components. The authors claim that Bi2Te3-polypyrrole exhibited the considerably high Seebeck coefficient compared to pristine Bi2Te3, and thus it recorded somewhat increased power factor despite the loss in electrical conductivity caused by the organic component, polypyrrole. In addition, the authors claim that the Bi2Te3-polypyrrole also exhibited much lower thermal conductivity than the pristine Bi2Te3 because of the phonon scattering effect at the interface. Finally, the authors claim that they successfully brought about the decoupling phenomenon of electrical and thermal properties by de- vising the inorganic/organic composite and adjusting its fabrication condition, thereby optimizing its thermoelectric performance, which is estimated as predominant property for n-type binary Bi2Te3 ever reported.
General comment: This manuscript is adequately written and organized. The methods also seem to be adequately described in detail. Nevertheless, to increase the quality and the impact of this work I suggest to split the current section "Results and discussion" in two different sections : "Results" and "Discussion". Indeed, in this way the authors could describe their results without comments within the "Results" section, while they can better describe the value of their work in comparison to the current state of the art within the "Discussion" section.
Some detailed comments:
3. Result and Discussion
*) This section should be split into two different sections "Results" and "Discussion", as already pointed out within the General comment. In particular, within the "Discussion" section, the authors should undeline the value of their work with respect to the current state of the art.
Lines: "4. Conclusions 290
Bi2Te3 was hybridized with polypyrrole to produce a bulk composite including the 291
interface, where the effects of energy band junction and phonon scattering effects possibly 292
occur; thus, the composite showed slight increase in the power factor while exhibiting the 293
significant reduction in thermal conductivity. This decoupling effect of the electrical and 294
thermal properties resulted in the remarkably enhanced ZT values, which may be evaluated as the highest performance among n-type binary Bi2Te3. We currently conduct further 296
studies on interaction mechanism and theory and various compositions between the components."
*) The authors should enlarge this section summarizing the novelty and the value of this work.
Round 2
Reviewer 1 Report
I recommend the paper for publication in the present form